# Gold endowments of porphyry deposits controlled by precipitation efficiency

Massimo Chiaradia[1]*

Porphyry deposits are natural suppliers of most copper and significant gold to our society. Whereas the Cu-richest (Au-poor) porphyries are related to Andean-type subduction and typical calc-alkaline magmatism, the Au-richest porphyries are associated with high-K calc-alkaline to alkaline magmatism in late to post-subduction or post-collision and extensional settings, and subordinately with calc-alkaline magmatism. The reasons behind these associations and the large variations in metal endowments of porphyry Cu–Au deposits remain obscure. Here, I show that porphyry Cu–Au deposits define two distinct trends in Au vs. Cu tonnage plots (Cu-rich and Au-rich). Metal endowments for both trends grow larger the longer the mineralization process. However, Au is precipitated at much higher rates in Au-rich than in Cu-rich porphyry deposits. Using Monte Carlo simulations of petrologic processes, I show that whereas Cu-rich porphyries require large amounts of magma and water to be formed, Au-rich porphyries are the result of a better efficiency of Au precipitation.

[1] Department of Earth Sciences, University of Geneva Rue des Maraîchers 13, 1205, Geneva, Switzerland. *email: Massimo.Chiaradia@unige.ch

Porphyry copper–gold deposits are large volume, low-grade disseminations formed by precipitation of copper and gold (plus molybdenum) from fluids of magmatic origin[1]. These deposits form at shallow crustal levels (mostly <5 km depth) in association with variably large magmatic reservoirs emplaced at 10–15 km depth feeding the shallower porphyritic fingers, which are the focus of the mineralisation[1]. Large magmatic reservoirs are in turn fed by deep (mid-to-lower crustal) magma accumulation zones[2].

The processes leading to the formation of these deposits, to their highly variable metal endowments (<1 to >100 Mt Cu and tens to thousands of tons Au) and to their association with rocks of different chemistry are complex, multi-step, and not well understood[3–5]. The majority of porphyry Cu–Au deposits are associated with typical calc-alkaline magmas (often high Sr/Y) in Andean-type subduction zones[1]. Recently it has become evident that Cu–Au deposits are also associated with lower Sr/Y, high-K calc-alkaline to alkaline magmas (e.g., Cadia, Bajo de la Alumbrera, Ok Tedi, Grasberg, Bingham and Kalmakyr)[6–10] (Supplementary Data 1) in complex late subduction, post-subduction and post-collision or extensional back-arc settings[6,8,11–14].

Here, I present a model to explain the variable Cu and Au endowments of porphyry Cu–Au deposits and their association with diverse magma types[8] and geodynamic settings[6,8,15]. I show that whereas the Cu-rich deposits require large volumes of hydrous magmas to be formed, Au-rich porphyry deposits form due to a better efficiency of Au precipitation.

## Results and discussion

**Data collection and filtering**. Metal endowments, rock geochemistry (Sr/Y values, magma affinity in terms of alkalinity) and geochronological data of 118 porphyry Cu–Au deposits (Supplementary Data 1) have been collected from previous studies and from online resources (USGS Porphyry Copper deposits of the world at http://mrdata.usgs.gov/porcu; http://www.portergeo.com.au/database/). Available Sr/Y values of magmatic rocks associated with each porphyry deposit were averaged and the associated 1 standard deviation values were calculated (Supplementary Data 1).

The magmatic affinity in terms of alkalinity of the magmatic rocks associated with the deposits was derived mostly from a previous study[7] and implemented by data from additional studies carried out on porphyry deposits which were not reported by ref. [7] (Supplementary Data 1). For the latter case magmatic affinity was evaluated using $K_2O$ enrichment in a $K_2O$ vs. $SiO_2$ plot[16], which allows the discrimination of rocks into calc-alkaline, high-K calc-alkaline and alkaline (shoshonitic). When geochemical analyses were not available, discrimination was done using the nomenclature of associated porphyritic rocks (see Methods for details).

The Cu and Au endowments here reported (Supplementary Data 1) are undoubtedly subject to uncertainty as shown by different values reported for the same deposit by distinct sources (Supplementary Data 1) and refinement of the reserves and resources through time. However, the overall range of the metal endowments of all world porphyries spans several orders of magnitude, which is much larger than possible metal endowment uncertainties of a single deposit.

Another point to highlight is that multi-stage deposits like Grasberg are characterised by individual ore bodies, formed at different times, which may have variable Cu/Au ratios. At Grasberg, mineralisation occurring within Dalam rocks and in the Ertsberg body has slightly higher Cu/Au ratios (~2.0; where Cu is in wt% and Au is in $g\,t^{-1}$) than all other ore bodies (0.75–1.40) and than the bulk Grasberg–Ertsberg district (~1.0)[14]. The reasons of these local differences are overprinting of subsequent ore stages and different depths of ore formation[14] (the higher Cu/Au values of ores both in the Dalam rocks and at Ertsberg are accompanied by higher contents of molybdenum in the deeper parts of the orebodies). In the following I have considered the Au and Cu endowments of the bulk Grasberg–Ertsberg district, which reflect Au/Cu ratios of the greatest majority (>90% in terms of tonnage)[14] of the ore bodies of the district.

The used geochronology data on porphyry deposits (Supplementary Data 1 and Supplementary Note 1) were obtained through state-of-the art techniques[17] (U-Pb dating of zircons of porphyry intrusions by CA-ID-TIMS, SHRIMP and LA-ICPMS, Re-Os ages of molybdenite by N-TIMS, $^{40}Ar/^{39}Ar$ dating of hydrothermal minerals: Supplementary Note 1) during the last 20 years and most of them (15 out of 22) during the last 10 years. The data, in conjunction with interpretations provided by the authors of these studies, were used to calculate the overall duration of the ore mineralisation process, i.e., the temporal interval encompassing, as far as possible, the bulk of the mineralising process at a specific deposit (Supplementary Data 1 and Supplementary Note 1). This was based on either temporal bracketing using pre-ore and syn- to post-ore porphyry U-Pb zircon dating, or Re-Os dating of molybdenite from multiple ore stages texturally constrained, eventually implemented by $^{40}Ar/^{39}Ar$ dating of alteration minerals associated with the ore (see Supplementary Note 1 for a detailed description of how the overall durations of the mineralisation process were obtained for each deposit). This is in particular true for the largest composite porphyry systems, like, among others, Chuquicamata, Rio Blanco and Grasberg. Because of unavoidable undersampling, the time intervals so determined are first order approximations of the real durations of the mineralising events at each porphyry deposit. It is nonetheless significant that similar values for the duration of ore processes have been obtained by distinct studies, when these are available on the same deposit (e.g., El Teniente, Grasberg and Chuquicamata; Supplementary Data 1).

**Petrologic modelling**. Monte Carlo modelling of petrologic processes (Methods and Table 1) has been applied to extract information on metallogenic processes able to explain the Cu and Au endowments of porphyry deposits and their timescales of formation (see above). I have used the mass balance and petrologic approach of ref. [2] to estimate magma volumes and amounts of fluid, Cu and Au exsolvable from these magmas as well as their $SiO_2$ contents (for details see Methods, Supplementary Figs. 1–5, and Supplementary Table 1). The magma masses and volumes are determined parameterising the thermodynamic conditions outlined by ref. [18] for the generation of melts in hot crustal zones. In the model, basaltic melt is injected into the crust at variable depths at a fixed typical long-term average rate of 5 mm year$^{-1}$ (ref. [18]) for a time interval ranging between 0 and 5 Ma. Depending on the depth at which injection occurs, residual melt from fractionation of the injected basalt will start to accumulate after a certain incubation time (Supplementary Fig. 1). The dependence of the incubation time on depth of injection is explained by the fact that host rock temperature increases with depth according to the geothermal gradient (20 °C km$^{-1}$ in the model[18]). Therefore, at deeper levels (i.e., hotter host rock temperatures) incubation times for initial residual melt formation will be shorter. At the same time, continuous injection of the basaltic melt will also result in an increase of the temperature of the host rocks, which, after a certain incubation time, that is different from the one of residual melt formation, might reach the solidus of these rocks with their consequent partial melting (crustal partial melt: Supplementary Fig. 1). The resulting melt from all this process is a composite hybrid melt deriving from the sum of the

**Table 1 Input parameter values used for the Monte Carlo simulations. Simulations are carried out for an injection rate of 5 mm year⁻¹ of a basaltic melt at 1200 °C through a disk of 7500 m radius[a] (equivalent to a magma flux of 0.0009 km³ year⁻¹), into a crust characterised by a geothermal gradient of 20 °C km⁻¹ (ref. [18]).**

| Input parameter | Value(s) |
|---|---|
| Time | Random between 0 and 5 Ma |
| Pressure (calc-alkaline systems) | Random between 0.15 and 0.9 GPa |
| Pressure (alkaline systems) | Random between 0.15 and 0.6 GPa |
| $H_2O$ in parent magma | Random between 2 and 4 wt% [49] |
| $H_2O$ in crustal rocks | Random between 0.2 and 1 wt% [2] |
| Fluid-melt partition coefficient of copper | Random between 2 and 100 [2] |
| Fluid-melt partition coefficient of gold | Random between 10 and 100 [27] |
| Gold content in calc-alkaline magmas | Random between 6 and 9 ppb [47] |
| Gold content in alkaline magmas | Random between 10 and 32 ppb [25] |
| Copper content in calc-alkaline and alkaline magmas | Constrained through $SiO_2$–Cu relationship of alkaline and calc-alkaline magmas [45,46] |
| Cu precipitation efficiency in Cu-rich calc-alkaline systems | 50% (Fig. 3a–c, Fig. 4a, b) |
| Cu precipitation efficiency in Au-rich calc-alkaline systems | 30% (Fig. 4b) |
| Cu precipitation efficiency in Au-rich alkaline systems | 50% (Figs. 3a, 4a) |
| Au precipitation efficiency in Cu-rich calc-alkaline systems | 50% (Figs. 2a, 3b, c); 0.67% (Fig. 4a, b) |
| Au precipitation efficiency in Au-rich calc-alkaline systems | 5% (Fig. 4b) |
| Au precipitation efficiency in Au-rich alkaline systems | 50% (Figs. 2b, 3a); 3.3% (Fig. 4a) |

[a]An average size for crustal magma chambers, typically ranging between 5000 and 10,000 m (ref. [50])

residual and crustal melts at any time since the onset of injection and at any depth at which basaltic injection occurs (Supplementary Fig. 1). Through time the amount of melt accumulated at any specific depth will increase as shown by Supplementary Fig. 1. Melt productivity at deeper crustal levels will be larger than at shallower crustal levels (Supplementary Fig. 1).

The amount of dissolved $H_2O$ in such hybrid melts accumulated at different crustal depths and after different accumulation times (i.e., time since the onset of the injection process) can be determined taking into account the $H_2O$ initial contents of the primitive basaltic melt and of the crustal rocks (Table 1), and the pressure and melt composition dependency of $H_2O$ solubility in silicate melts[19] (Supplementary Figs. 2–4). Finally, the amounts of Cu and Au in the exsolvable $H_2O$ are determined by using a range of appropriate partition coefficients for these metals between fluid and silicate melt and appropriate Cu and Au contents in the melts (Table 1 and Supplementary Fig. 5).

**Metal endowments and timescales of Cu-Au porphyry deposits.** The plot of Au vs. Cu endowments shows that porphyry Cu–Au deposits define either a Cu-rich (Au/Cu ~$4 \times 10^{-6}$) or an Au-rich (Au/Cu ~$80 \times 10^{-6}$) trend (Fig. 1a). The Au-rich trend is essentially controlled by the seven largest gold deposits (containing almost the 60% of the gold of porphyry Cu–Au deposits[20]). These seven deposits (Kadjaran, Cadia, Kalmakyr, Oyu Tolgoi, Bingham, Grasberg and Pebble) are all associated with high-K calc-alkaline or alkaline rocks. Along the Au-rich trend there are also all other smaller deposits associated with variably alkaline magmas and several deposits associated with normal calc-alkaline magmas (e.g., Far Southeast-Lepanto, Reko Diq, Panguna, Cerro Casale, Batu Hijau to mention some of the largest ones). In contrast, all deposits of the Cu-rich trend are associated only with normal calc-alkaline rocks.

The two distinct trends are also recognisable in a plot of Au endowments vs. the durations of the ore formation process of the porphyry Cu–Au deposits (Fig. 1b): in the Cu-rich deposit trend Au is precipitated at a much slower average rate (~100 tons Au/ Ma) than in the Au-rich deposit trend (~4500 tons Au/Ma). The Au-rich trend is controlled by three large Au-rich porphyry deposits (for which robust geochronologic data are available), which are all associated with high-K calc-alkaline to alkaline

rocks (Grasberg, Bingham, Pebble) and by three Au-rich deposits associated with calc-alkaline rocks (Reko Diq, Far Southeast-Lepanto and Batu Hijau). All smaller sized Au-rich porphyries associated with variably alkaline rocks and several ones associated with normal calc-alkaline rocks fall on the Au-rich trend. Again, the Cu-rich trend is defined by deposits associated only with typical calc-alkaline rocks. Magmatic rocks associated with Au-rich porphyry deposits are characterised by lower Sr/Y values (~50 for the largest porphyry Au deposits)[21] compared to rocks associated with Cu-rich porphyry deposits ($100 \pm 50$)[2] (Fig. 1c and Supplementary Data 1).

**Possible causes of different Cu and Au endowments.** Chiaradia and Caricchi[2] suggested that the Cu endowment of Andean-type Cu-rich porphyry deposits is controlled by two main parameters: the volume of magma generated at mid-lower crustal depths, which determines the maximum amount of deliverable Cu, and the overall time interval during which magma, with its fluid and copper cargo, is transferred to shallower levels where fluid exsolution occurs and Cu is precipitated. The most favourable conditions to build the appropriately large volumes of magmas and fluids occur, as said above, in the middle to lower crust, where modelled magmas return Sr/Y values (50–150) that are in the same range as those of magmas associated with the largest porphyry Cu deposits[2]. The broad linear correlation between Cu endowments and durations of ore deposit formation[2] (Fig. 1d) suggests that the process of magma, fluid, and copper transfer to shallower levels occurs at a similar average rate for all Cu-rich deposits and that its duration is the main parameter controlling the Cu endowments in these deposits. A similar conclusion has also been reached by ref. [22]. In the Cu endowment vs. duration of ore deposit formation plot (Fig. 1d), Au-rich deposits fall towards the lower end of the same regression trend as the Cu-rich deposits, suggesting that Cu endowment controls and Cu precipitation efficiency are similar for both Cu-rich and Au-rich deposit types.

In contrast, the occurrence of two distinct linear trends in the Au–Cu tonnage and Au tonnage-ore duration plots (Fig. 1a, b) suggests that gold endowment is controlled by distinct processes in Cu-rich vs. Au-rich deposits. The association of the seven largest Au-rich porphyry deposits with mildly alkaline to alkaline rocks (Fig. 1a) could suggest some sort of petrogenetic control,

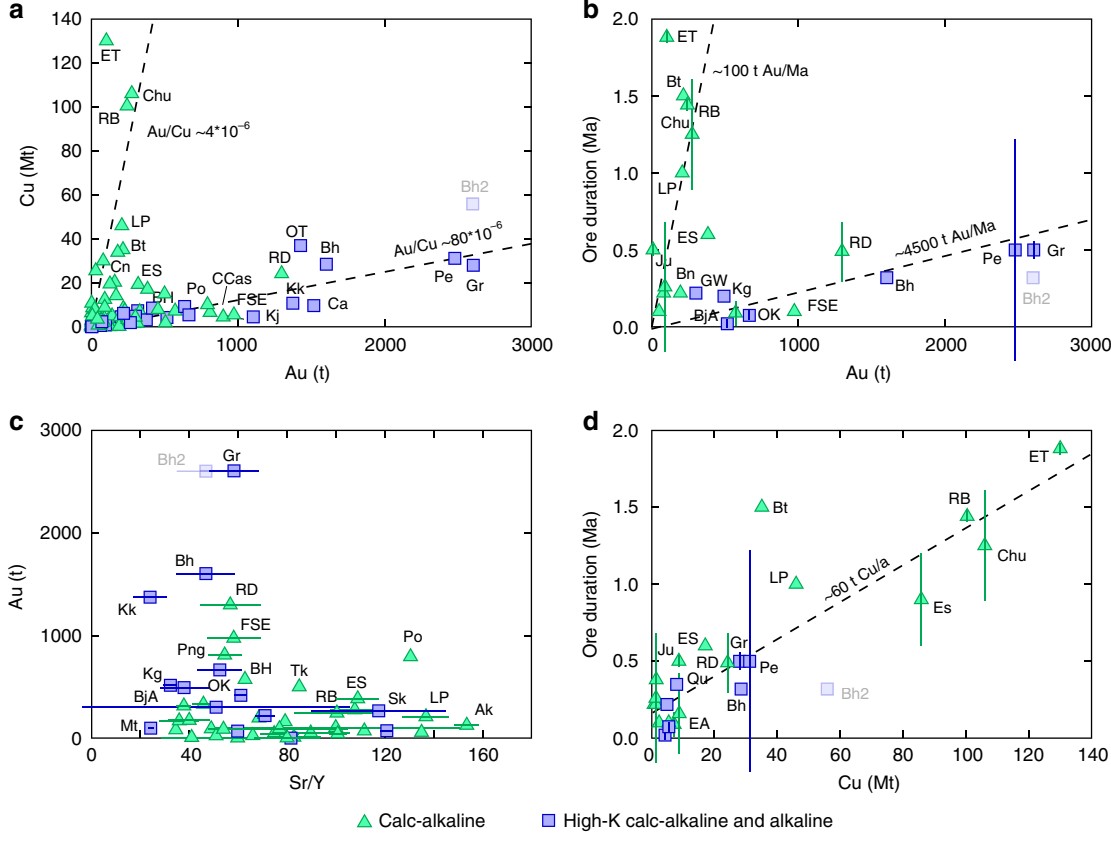

**Fig. 1 Metal endowments, geochemistry of associated rocks and ore durations of Cu-rich vs. Au-rich porphyry deposits. a** Cu (Mt) vs. Au (tons) of porphyry Cu–Au deposits; **b** ore duration (Ma) vs. Au (tons) of porphyry Cu–Au deposits. All deposits are roughly distributed along either one or the other of the two dashed lines allowing the identification of two distinct families of porphyry Cu–Au, the Cu-rich (Au/Cu~4 × 10$^{-6}$ and ~100 t Au Ma$^{-1}$) and the Au-rich (Au/Cu~80 × 10$^{-6}$ and ~4500 t Au Ma$^{-1}$). The dashed lines represent average rates of Cu and Au deposition and are not statistically best-fitted lines; **c** Au (tons) vs. Sr/Y average values of magmatic rocks associated with porphyry Cu–Au deposits. The bars for Sr/Y values are 1 s.d. uncertainties calculated from the available Sr/Y values of magmatic rocks associated with each deposit (see Supplementary Data 1); **d** ore duration (Ma) vs. Cu (Mt) of porphyry Cu–Au deposits. The bars associated with the ore duration values are propagated 2 s.d. uncertainties as explained in Supplementary Note 1. Abbreviations of porphyry deposits: Ak Aksug, BH Batu Hijau, Bh Bingham, BjA Bajo de la Alumbrera, Bt Butte, Ca Cadia, CCas Cerro Casale, Chu Chuquicamata, Cn Cananea, EA El Abra, ES El Salvador, ET El Teniente, FSE Far Southeast-Lepanto, Gr Grasberg, Ju Junin, Kg Kisladag, Kj Kadjaran, Kk Kalmakyr, LP Los Pelambres, Mt Marte, OK Ok Tedi, OT Oyu Tolgoi, Pe Pebble, Png Panguna, Po Potrerillos, Qu Qulong, RB Rio Blanco, RD Reko Diq, Sk Skouries, Tk Toki. Bingham has two points (Bh and Bh2) due to different tonnages reported in different studies (see Supplementary Data 1).

which is not clearly understood[8,23,24]. On the other hand, Au-rich porphyry deposits with variably large gold endowments are also associated with normal calc-alkaline magmatic rocks (Fig. 1a). This suggests that magma chemistry cannot be the only control on the formation of the Au-rich porphyry trend. Here, I explore three major mechanisms that could be responsible for the formation of Au-rich porphyry deposits and their preferential, but not unique, association with variably alkaline magmas: (i) higher Au contents in alkaline magmas[25] (and in calc-alkaline magmas associated with Au-rich porphyries), (ii) varying fluid-melt partition coefficient ($K_D$) values of Au between fluids and melts and (iii) different precipitation efficiencies.

**A precipitation efficiency control for Au endowments.** Monte Carlo simulations show that, assuming a commonly used 50% precipitation efficiency for both Au and Cu, magma volumes (~2000 km³) corresponding to the highest enrichments in copper (~100 Mt Cu) associated with calc-alkaline magmas[2] would provide Au in great excess (median value of ~14,000 tons Au) to the maximum gold endowment (~2700 tons Au) of Au-rich porphyry deposits (Fig. 2c; even higher potential Au endowments are associated with the largest simulated magma volumes of alkaline systems at 50% efficiency: Fig. 2d). This suggests that the

decoupling between Cu and Au endowments in Cu-rich vs. Au-rich deposits is unlikely to be related only to Au enrichment in alkaline magmas compared to calc-alkaline magmas[25,26], because the latter can exsolve fluids with largely enough gold to form the largest Au-rich porphyry deposits. Neither can varying fluid-melt $K_D$ values of Au explain the depleted Au contents of Cu-rich deposits. Indeed, using ranges of common fluid-melt $K_D$ values for Au (10–100)[27] and Cu (2–100)[2] and precipitation efficiencies of 50% for both Au and Cu, Monte Carlo simulations result in fluids with Au/Cu values much higher than those recorded by natural Au-rich porphyry deposits for both calc-alkaline and alkaline magmas (Fig. 3a, b). It is impossible to reproduce the low Au/Cu values of Cu-rich deposits unless unreasonably low fluid-melt $K_D$ values for Au («1) are assumed (Fig. 3c). Additionally, modelled and geological fluids exsolved from magmas (volcanic emissions and single-phase fluids of porphyry deposits) have very similar Au/Cu values (Fig. 3, Table 2). This supports the contention that Au and Cu in magma-derived fluids occur in concentrations that are in agreement with those obtained using experimentally determined fluid-melt $K_D$ values of Au and Cu.

The plots of Figs. 2 and 3, thus, suggest that the different Au/Cu trends of Au-rich and Cu-rich deposits could be due to different Au precipitation efficiencies. Indeed, the two porphyry

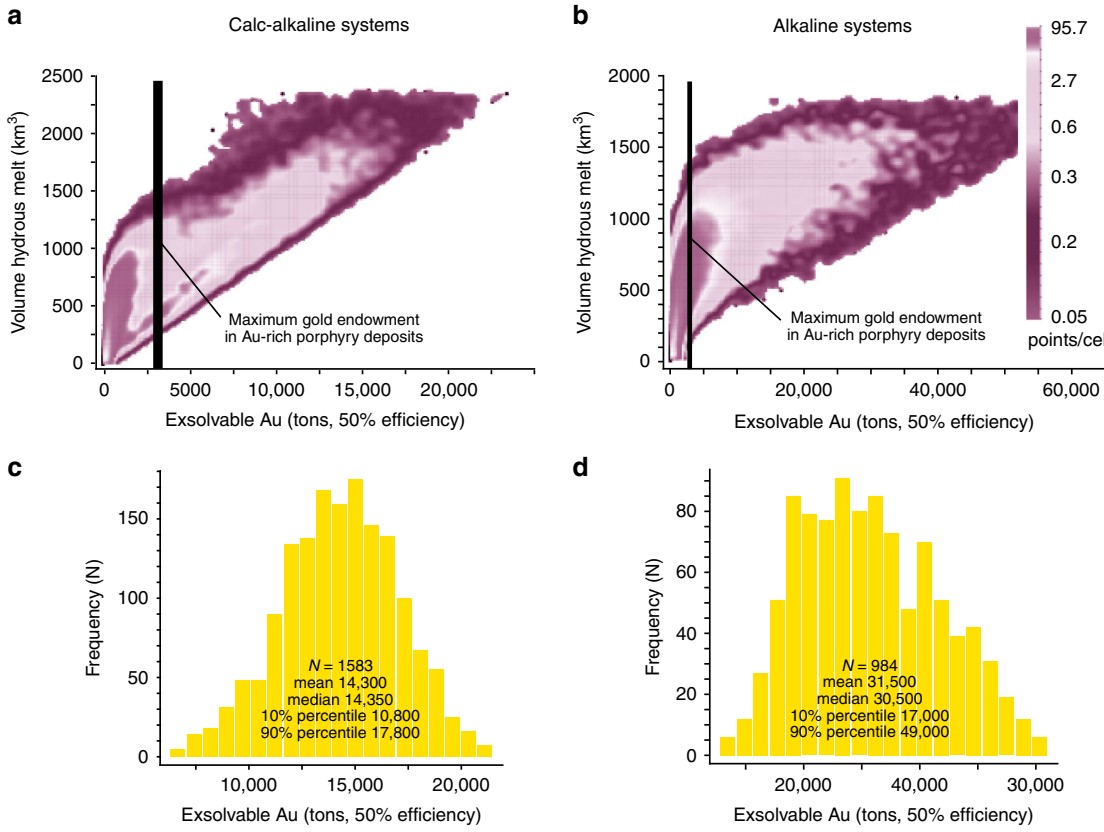

**Fig. 2 Monte Carlo simulations of exsolvable gold and relationship to magma volumes.** Density plots of the Monte Carlo simulations (>25,000) show that both calc-alkaline (**a**) and alkaline (**b**) magmatic systems can potentially exsolve fluids, during the lifetimes of the porphyry deposits indicated by geochronology, carrying and precipitating (50% efficiency) much larger amounts of gold than those recorded by the Au-richest porphyry deposits; **c** histogram of the exsolvable gold (50% efficiency) for the magma volume range (1750–2250 km$^3$) associated with the largest Cu endowments (50% precipitation efficiency) in calc-alkaline magmatic systems: this corresponds to the density distribution of exsolvable Au simulations for the volume interval 1750–2250 km$^3$ in (**a**); **d** histogram of exsolvable gold (50% efficiency) for the largest magma volume range (1000–1500 km$^3$) obtained in the simulations for alkaline magmas.

trends in the Au–Cu space are well reproduced by Monte Carlo simulations carried out for gold precipitation efficiencies that are lower than those of Cu by a factor of ~6–15 in Au-rich deposits and by a factor of ~75 in Cu-rich deposits (Fig. 4a, b; see Methods for more details). This translates into Au precipitation efficiencies that are ~5–12 times higher in Au-rich porphyries than in Cu-rich porphyries (Fig. 4a, b and Methods).

An increased precipitation efficiency of Au, resulting in the Au-rich trend of porphyry deposits, could be due to: (i) the shallower depth at which Au-rich deposits form[28] and (ii) the higher stability of hydrosulphide gold in alkali-rich fluids[24]. As discussed in detail by ref. [28], in shallow porphyry systems (<~3 km) gold and copper solubility decreases rapidly in an expanding S-rich vapour, which carries both metals. The result is the co-precipitation of Cu and Au and high Au/Cu values. In contrast, in deeper porphyry systems (>~3 km) a single-phase fluid pre-dominates from which mostly Cu precipitates upon cooling, whereas Au remains in solution in a dense vapour phase. According to ref. [24] the presence of alkali chlorides strongly increases the solubility of gold in H$_2$S-bearing fluids and could explain the association of Au-rich porphyry deposits with alkaline magmas, from which, supposedly, fluids with higher contents of alkali chlorides are exsolved.

An additional factor responsible for gold and copper decoupling in some specific porphyry deposits could be the reduced nature of the magmatic-hydrothermal system, either inherent to the magma or resulting from interaction of the fluids

with reduced host rocks[29]. Different from Cu, which solubility decreases in reduced ore fluids, gold can be transported at similar concentrations by ore fluids independent of their oxidation state[29]. Therefore, it has been suggested that reduced magmatic-hydrothermal systems could be responsible for the formation of some Au-rich porphyry deposits[30].

**A tectonic control for Cu vs. Au endowments.** In Andean-type subduction arcs, long-lived periods of compression (>2 Ma) lead to the accumulation of variably large magma volumes at deep crustal levels with a typical calc-alkaline signature hallmarked by high Sr/Y values[2,31]. In such a context, porphyry Cu-rich deposits form because they essentially depend on large magma volumes accumulated at mid-lower crustal depths during the compressional period, and on the subsequent duration of magmatic-hydrothermal leakage of the deep reservoir to the shallower crust, where ore deposition occurs. During this process, gold is precipitated at a low average rate (Fig. 1b) because fluids exsolved from these calc-alkaline magmas have poor precipitation efficiencies for gold (~75 times less than Cu precipitation efficiency: see above). This is likely due to an average deep formation of Cu-rich deposits in such a context[28] and, perhaps, to inefficient chemistry of associated fluids[24]. Figure 5a, b shows that the largest Cu-rich porphyry deposits (>30 Mt Cu) associated with calc-alkaline magmas occur at depths >~3 km and have gold endowments of <500 tons Au. In this case, a significant amount of gold

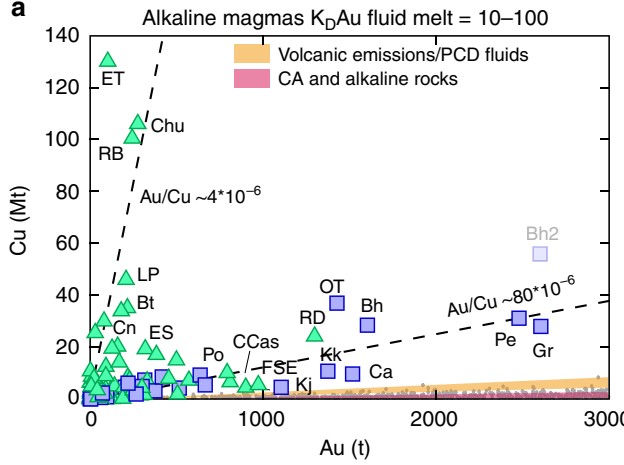

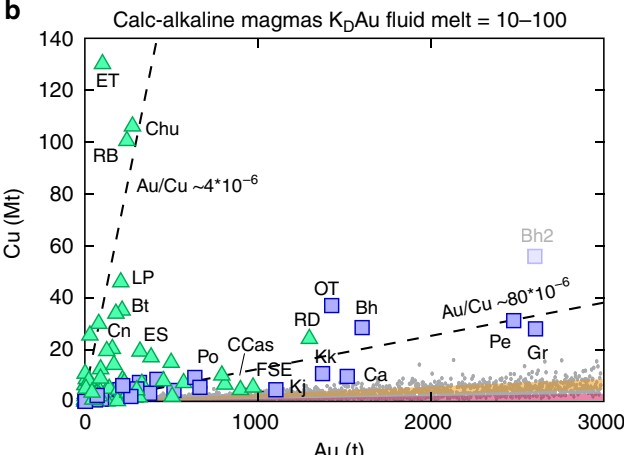

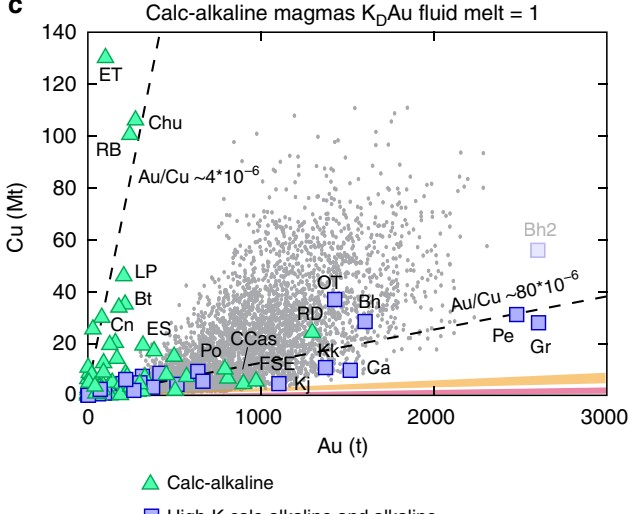

**Fig. 3 Monte Carlo simulations (grey dots; >7000) of the Au and Cu endowments obtained for different fluid-melt $K_D$ values of Au.** Using a range of commonly accepted $K_D$ values for Au (10–100)[27] and Cu (2–100)[2] results in exsolved fluids with extremely high Au/Cu values both for alkaline (**a**) and calc-alkaline (**b**) magmas. The very low Au/Cu values of Cu-rich deposits can only be obtained by assuming unrealistically low $K_D$ values («1) for Au (**c**). Also shown are the Au/Cu ratios for volcanic emissions and single-phase fluids of porphyry deposits (orange field: Table 2) and of calc-alkaline and alkaline rocks (red field: Table 2). Abbreviations of porphyry deposits: Bh Bingham, Bt Butte, Ca Cadia, CCas Cerro Casale, Chu Chuquicamata, Cn Cananea, ET El Teniente, FSE Far Southeast-Lepanto, Gr Grasberg, Kj Kadjaran, Kk Kalmakyr, LP Los Pelambres, OT Oyu Tolgoi, Pe Pebble, Po Potrerillos, RB Rio Blanco, RD Reko Diq. Bingham has two points (Bh and Bh2) due to different tonnages reported in different studies (see Supplementary Data 1).

of magma evolution[31]. The overall lower Sr/Y values (~50) of variably alkaline (and some calc-alkaline) magmas associated with Au-rich porphyry deposits (Fig. 1c) support their evolution at average shallower crustal levels because Sr/Y is a proxy for the depth of magma evolution[31,35,36]. All Au-rich porphyry deposits associated with variably alkaline magmas are indeed formed at shallow crustal levels (<~3 km; Fig. 5b), most likely due to the association of these magmas with tectonic (extension) and geodynamic (thinner crust) contexts that favour their emplacement at shallow crustal levels.

In contrast, porphyry deposits associated with calc-alkaline magmatic rocks encompass a broader range of depths of formation, but only the shallow (<~3 km) systems may be associated with large (>500–<1500 tons of Au) Au-rich porphyries (Fig. 5b). This suggests that shallow level magma emplacement and consequent formation of Au-rich systems may also occur in association with calc-alkaline magmas both in Andean-type subduction settings (e.g., Maricunga Au-rich porphyry systems[37]), for example during extensional periods intercalated within an overall compressional regime[38,39], and in crustally thinner island arc settings, for example during arc-parallel extension associated with collision (e.g., Batu Hijau[40] and Grasberg[14]).

In support to the above arguments, Sr/Y average values of both calc-alkaline and variably alkaline magmatic systems (which are a proxy for the average depth of magma evolution: see above) correlate with the depth of porphyry formation (Fig. 5c; the only exception is Chino-Santa Rita): in other words, the shallower or deeper is the average magma evolution in the crust (independent from magma chemistry), the shallower or deeper is the emplacement of magma in the upper crust and consequent porphyry formation. Most likely, this is a consequence of both these processes being controlled by crustal thickness and tectonic regime (compression vs. extension).

The distinct association of the largest Au-rich porphyry deposits with mildly alkaline to alkaline magmatic rocks (Fig. 1a), nonetheless, calls for additional factors that further enhance their gold endowment. A comparison of Monte Carlo modelling for Au-rich alkaline and calc-alkaline systems (Fig. 4a, b) suggests that the higher gold endowments of Au-rich porphyry deposits associated with alkaline magmas can be explained by the higher gold contents in the alkaline magmas. Another factor enhancing the Au endowments of Au-rich porphyry deposits associated with alkaline systems could be the favourable chemistry of the fluids associated with such magmas[24].

On the other hand, shallow crust magma evolution is not favourable for the generation of the largest possible magma volumes and Cu endowments[2,33]. Consequently, shallowly formed deposits cannot reach the most outstanding Cu

may be lost to volcanic emissions, which have similarly high Au/Cu values to those of the magmatic rocks and to those of single-phase fluids exsolved from magmas at high pressure (Fig. 5a, b).

In late to post-subduction and post-collision settings, mildly alkaline to alkaline magmas are associated with extension[6] or with arc reversal in thinner island arcs[26] (e.g., Grasberg[13,14], Bingham[11] and Kisladag[32]). Extension favours the ascent, evolution and emplacement of magmas to shallower crustal levels[33,34] whereas thinner crust results in shallower average levels

**Table 2 Au/Cu values of the porphyry trends and different geological materials.**

| Typology | Au/Cu | Reference/Remarks |
|---|---|---|
| Au-rich deposit trend | ~0.000080 | This work |
| Cu-rich deposits trend | ~0.000004 | This work |
| Volcanic emissions | 0.00033 | Median value of five volcanoes from refs. [51,52] |
| Calc-alkaline rocks | 0.0009 | Median value of ~1000 Monte Carlo simulations from this work using Cu and Au data from refs. [2,47] |
| Alkaline rocks | 0.0034 | Median value of ~1000 Monte Carlo simulations from this work using Cu and Au data from refs. [2,25] |
| Low salinity single-phase high pressure fluid inclusions from porphyry deposits | 0.0006 | From ref. [53] |

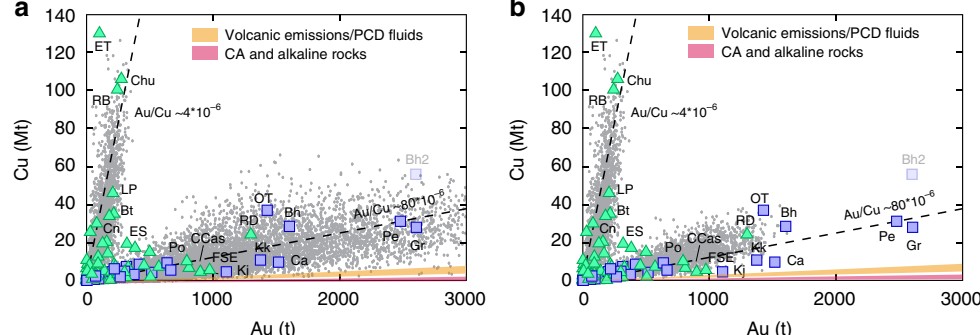

**Fig. 4 Monte Carlo simulations (light grey dots; _N_ > 20,000) for the trends of Cu-rich and Au-rich porphyry deposits. a** Simulations carried out for Au precipitation efficiencies that are approximately five times higher for the alkaline systems-related Au-rich trend than for the calc-alkaline systems-related Cu-rich trend (see text, Methods, and Table 1); **b** simulations carried out for Au precipitation efficiencies that are ~12 times higher for the calc-alkaline systems-related Au-rich trend than for the calc-alkaline systems-related Cu-rich trend (see text, Methods, and Table 1). Also shown are the Au/Cu ratios for volcanic emissions and single-phase fluids of porphyry deposits (orange field: Table 2) and of calc-alkaline and alkaline rocks (red field: Table 2). Abbreviations of porphyry deposits: Bh Bingham, Bt Butte, Ca Cadia, CCas Cerro Casale, Chu Chuquicamata, Cn Cananea, ET El Teniente, FSE Far Southeast-Lepanto, Gr Grasberg, Kj Kadjaran, Kk Kalmakyr, LP Los Pelambres, OT Oyu Tolgoi, Pe Pebble, Po Potrerillos, RB Rio Blanco, RD Reko Diq. Bingham has two points (Bh and Bh2) due to different tonnages reported in different studies (see Supplementary Data 1).

endowments (>50 Mt Cu) of magmatic systems associated with typical Andean-type subduction under thick continental crust (Fig. 1d).

**A multi-step process for Cu–Au endowments.** Whereas depth of porphyry formation and chemistry of magmas and associated fluids seem to control the Au-rich vs. Cu-rich nature of porphyry Cu–Au deposits, the increases of the Cu and Au endowments with ore deposition duration (Fig. 1b, d) suggest that the final Cu and Au endowments of these deposits are determined by the cumulative number of mineralising steps[41,42] that are ultimately controlled by magma volume and ore process duration[2]. The difference is that variably alkaline systems and shallow crustal calc-alkaline systems are inherently associated with magmas, whose fluids are tectonically (i.e., shallow emplacement: ref.[28]) and chemically[24] optimised for high gold precipitation efficiency. In contrast, typical calc-alkaline (high Sr/Y) magmas form in a geodynamic context that favours enormous magma accumulations, which are necessary to produce behemothian Cu-rich deposits[2], but are emplaced at depths at which the exsolved fluids are less efficient for gold precipitation.

## Methods

The petrologic model here used is a reduced version (less computed output parameters) of the model used by Chiaradia and Caricchi[2]. It consists of a set of equations written in Excel (Supplementary Data 2, Supplementary Note 2, Supplementary Tables 2–6) to quantify, using a Monte Carlo approach (Table 1), the following main parameters: (i) the amounts of hybrid melt produced in the crust (melt productivity indicates the amount of hybrid melt accumulated divided by the amount of total intruded basaltic melt) through processes typical of hot zones[18] as

discussed in the main text; (ii) their water contents (in solution, in excess and exsolvable at pressure of saturation) and (iii) the Cu and Au contents in the exsolvable water at the pressure of saturation and the $SiO_2$ composition of the hybrid melts produced within the crust. In the specific case several thousands of Monte Carlo simulations were used to obtain the model results discussed in the text.

**Melt productivity.** Melt productivity is quantified at different crustal depths under a typical average arc magma flux (5 mm year$^{-1}$ of basaltic melt injection rate through a circular section of 7500 m of radius, equivalent to 0.0009 km$^3$ year$^{-1}$), using the model of ref.[18], for time intervals between 0 and 5 Ma, and for pressures between 0.15–0.9 GPa (corresponding to crustal depths of ~5–~30 km) for calc-alkaline systems and between 0.15 and 0.6 GPa (corresponding to crustal depths of ~5–~20 km) for alkaline systems. Different pressure intervals for calc-alkaline and alkaline systems are used because the evolution and emplacement of variably alkaline magmas occurs in extensional settings and/or in thinner crust settings (see above) at shallower average levels than typical calc-alkaline systems (Figs. 1c, 5c). A pressure range between 0.15 and 0.6 GPa has also been used to simulate Au-rich porphyry deposits associated with calc-alkaline magmas (see main text). Using the same pressure interval for both calc-alkaline and alkaline systems (0.15–0.9 GPa) does not change significantly the simulations.

Pressure (P) and time (T) of maturation of the magmatic systems are allowed to vary randomly within the above mentioned fixed limits (Table 1) to obtain, for any random value of P and T, the corresponding value of melt productivity using a Monte Carlo method.

The curves of melt productivity have been parameterised from ref.[18] for both residual ($M_{residual}$) and crustal melt ($M_{crustal}$) fractions (Supplementary Fig. 1), which are expressed as polynomial functions of pressure (P) of the type

$$M_{residual} = xP^2 + yP + z, \qquad (1)$$

$$M_{crustal} = xP^2 + yP + z, \qquad (2)$$

where M is the residual or crustal melt fraction, P is the pressure at which injection and accumulation of residual or crustal melt is occurring and x, y and z are variables that depend on the incubation time through best fit polynomial equations

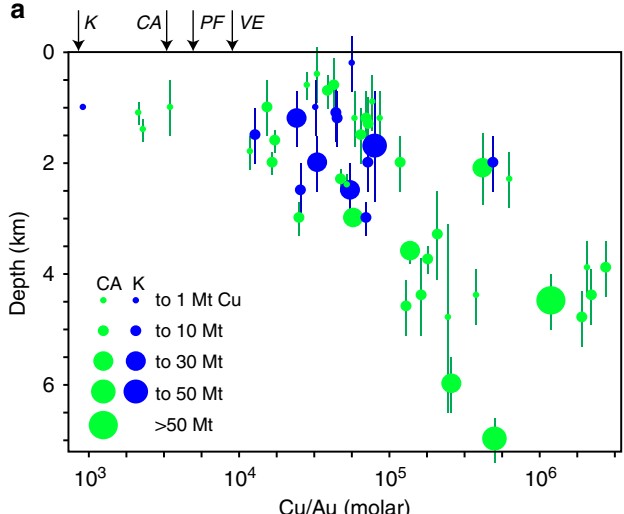

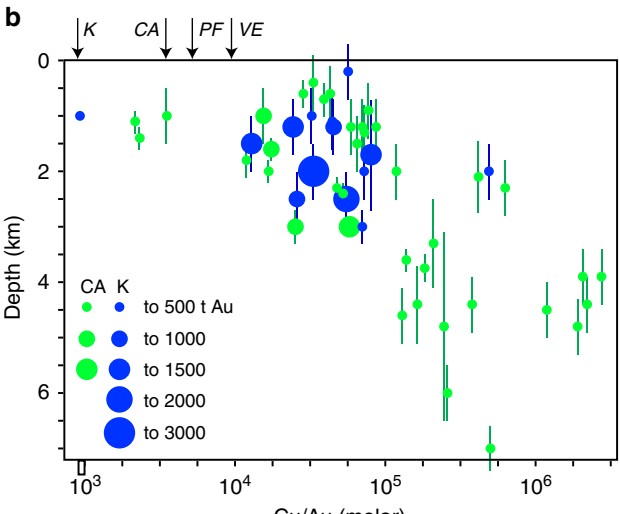

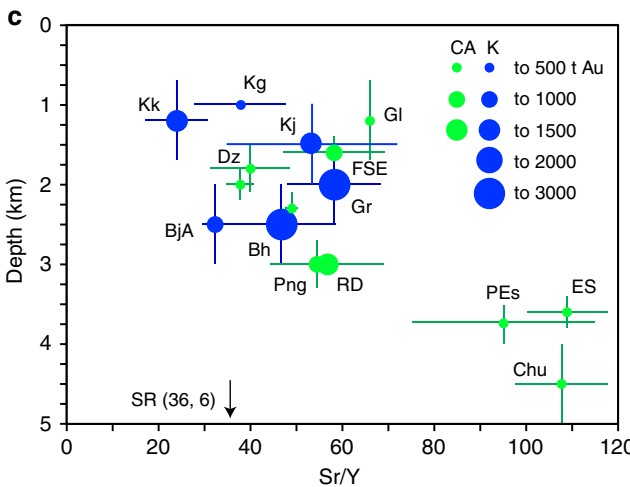

**Fig. 5 Depth of formation vs. Cu/Au molar ratios of porphyry Cu-Au deposits and Sr/Y average values of associated magmatic rocks.** The size of the symbols corresponds to different copper (**a**) and gold (**b**, **c**) tonnages as indicated in the legend. Green and blue colours of the symbols refer respectively to calc-alkaline (CA) and high-K calc-alkaline to alkaline (K) magmatic systems. The error bars associated with depth values are from ref. [28] (Supplementary Data 1). The bars for Sr/Y values are 1 s.d. uncertainties calculated from the available Sr/Y values of magmatic rocks associated with each deposit (see Supplementary Data 1). Abbreviations of porphyry deposits: Bh Bingham, BjA Bajo de la Alumbrera, Chu Chuquicamata, Dz Dizon, ES El Salvador, FSE Far Southeast-Lepanto, Gl Granisle, Gr Grasberg, Kg Kisladag, Kj Kadjaran, Kk Kalmakyr, PEs Pampa Escondida, Png Panguna, RD Reko Diq, SR Chino/Santa Rita. Other abbreviations: K = Cu/Au moral ratio of alkaline rocks; CA = Cu/Au moral ratio of calc-alkaline rocks; PF = Cu/Au moral ratio of single-phase porphyry fluids; VE = Cu/Au moral ratio of volcanic emissions (Table 2).

constant values different for each one of the $x$, $y$ and $z$ variables (see Supplementary Data 2).

**H₂O concentrations in the hybrid melt**. Melt productivity as determined above was coupled to $H_2O$ concentrations in the hybrid melt assuming a geologically sound[43] range of initial $H_2O$ contents in the mantle-derived basalt (2–4 wt%) and in the amphibolitic (lower) to greywacke (upper) crust (0.2–1.0 wt%) (Table 1) and assuming a completely incompatible behaviour of $H_2O$ during the hybrid melt accumulation process. VolatileCalc[19] was then used to calculate water solubility in melts according to pressure of accumulation and hybrid melt composition (Supplementary Fig. 2). This way, the $H_2O$ contents of the melts and the degree of $H_2O$ over- or under-saturation in the hybrid melts produced at different crustal levels (P) and after different durations of injection could be determined. This allowed the determination of the amount of exsolvable $H_2O$ (i.e., dissolved in undersaturated magmas) associated with any specific hybrid melt produced after any injection time, at any crustal depth. At any specified pressure, $H_2O$ solubility is linked to melt fraction (M) by a best-fit polynomial equation (Supplementary Fig. 3) of the type

$$H_2O = rM^2 + sM + t, \qquad (6)$$

where $H_2O$ is in wt%, $M$ = melt fraction and $r$, $s$, $t$ are pressure-dependent variables according to other second order polynomial equations (Supplementary Fig. 4) of the type

$$r = h'P^2 + k'P + i', \qquad (7)$$

$$s = h''P^2 + k''P + i'', \qquad (8)$$

$$t = h'''P^2 + k'''P + i''', \qquad (9)$$

where $P$ is the pressure and $h'$, $h''$, $h'''$, $k'$, $k''$, $k'''$, $i'$, $i''$, $i'''$ are constant values specific to each one of the $r$, $s$, $t$ variables (Supplementary Fig. 4 and Supplementary Data 2). Combining these equations allows reproduction of the solubility of $H_2O$ for any random $P$ and $M$ value used in the Monte Carlo simulations.

**Hybrid melt SiO₂ content**. In order to link the melt productivity of the model of ref. [18] to the $SiO_2$ content I used the relationship below (Supplementary Table 1) between melt fraction (M) and $SiO_2$ (in wt%), based on the mid-values of $SiO_2$ for the fields of basalt, basaltic andesite, andesite, dacite and rhyolite of the Total Alkali-Silica diagram[44] and the mid-values of the melt fraction and the corresponding composition attributed by ref. [18] (e.g., Fig. 8 of ref. [18]).

In a bivariate plot, the two variables above are linked through the equation

$$SiO_2 = 35.43629M^2 - 68.8591M + 82.43897. \qquad (10)$$

The modelled $SiO_2$ composition is used to determine the Cu contents in the modelled melts, as explained below.

**Amounts of Cu in the exsolvable fluid**. The amounts of Cu in the exsolvable fluid depend on the concentration of Cu in such a fluid. The latter depends on the Cu concentration in the melt and on the value of the fluid-melt partition coefficient, which determines how much Cu goes into the exsolvable fluid once it separates from the melt. For Cu concentrations in the hybrid melt the $SiO_2$-dependent Cu concentrations of continental arc magmas of ref. [45] was used. The $SiO_2$–Cu relationship is best fitted by a second order equation

$$y = 0.0632x^2 - 10.118x + 407.63, \qquad (11)$$

where $y$ = Cu (ppm) and $x$ = $SiO_2$ (wt%). This equation expresses the covariation between median Cu and $SiO_2$ values from thick arc magmas (>30 km)

of the type

$$x = a'T^6 + b'T^5 + c'T^4 + d'T^3 + e'T^2 + f'T + g', \qquad (3)$$

$$y = a''T^6 + b''T^5 + c''T^4 + d''T^3 + e''T^2 + f''T + g'', \qquad (4)$$

$$z = a'''T^6 + b'''T^5 + c'''T^4 + d'''T^3 + e'''T^2 + f'''T + g''', \qquad (5)$$

where $T$ is the incubation time since the onset of the injection of basaltic magma and $a'$, $a''$, $a'''$, $b'$, $b''$, $b'''$, $c'$, $c''$, $c'''$, $d'$, $d''$, $d'''$, $e'$, $e''$, $e'''$, $f$, $f'$, $f''$, $g'$, $g''$, $g'''$ are

(Supplementary Fig. 5). Since there is some scatter in the $SiO_2$–Cu relationship, all possible values within the upper and lower boundaries of this scatter have been considered and implemented in the Monte Carlo modelling. Georgatou and Chiaradia[46] have shown that also high-K calc-alkaline, shoshonitic and other alkaline rocks follow a similar trend in the $SiO_2$–Cu plot. Therefore, the Cu contents of melts in the model were taken to be the same for calc-alkaline and variably alkaline rocks. A conservative random variation of Cu fluid-melt $K_D$ values between 2 and 100[2] (Table 1) was used for all types of hybrid melt produced.

**Amounts of Au in the exsolvable fluid**. As for Cu, also the amounts of Au in the exsolvable fluid depend on the concentration of Au in such a fluid. The latter depends on the Au concentration in the melt and on the value of the fluid-melt partition coefficient, which determines how much Au goes into the fluid once it separates from the melt (Table 1). For the fluid-melt partition coefficient of Au a range of values between 10 and 100[27] was used.

Different Au concentrations were used for calc-alkaline magmas of intermediate compositions (6–9 ppb)[47] and alkaline magmas (10–32 ppb)[25] (Table 1). In the model the concentrations were allowed to vary randomly between the above ranges for the two distinct magma types.

**Precipitation efficiency**. The total amounts of Cu and Au in the exsolvable fluid were reduced according to the precipitation efficiencies discussed in the text and reported in Table 1 to simulate the amounts of metals effectively precipitated in the ore deposit.

Monte Carlo modelling in Figs. 2 and 3 is carried out with Au and Cu precipitation efficiencies of 50%. Monte Carlo modelling in Fig. 4a is carried out assigning a fixed Cu precipitation efficiency of 50% and variable Au precipitation efficiencies of 0.67% and 3.33% to the calc-alkaline Cu-rich and alkaline Au-rich magmatic systems, respectively. This corresponds to Au precipitation efficiencies that are ~75 and ~15 times lower than Cu precipitation efficiency in calc-alkaline and alkaline systems, respectively, which translates into a factor of ~5 more efficient Au precipitation in alkaline than in calc-alkaline magmatic systems.

Monte Carlo modelling in Fig. 4b is carried out assigning Cu precipitation efficiencies of 50% to the Cu-rich calc-alkaline systems and of 30% to the Au-rich calc-alkaline systems. Au precipitation efficiencies of 0.67% and 5% were used for the calc-alkaline Cu-rich and calc-alkaline Au-rich magmatic systems, respectively, to obtain the best fit of the simulations with the Au-rich deposits associated with calc-alkaline magmas. This corresponds to Au precipitation efficiencies that are ~75 and ~6 times lower than Cu precipitation efficiency in calc-alkaline Cu-rich systems and calc-alkaline Au-rich systems, respectively, which translates into a factor of ~12 more efficient Au precipitation in Au-rich calc-alkaline than in Cu-rich calc-alkaline systems.

**Magmatic affinity (calc-alkaline vs. high-K calc-alkaline to alkaline)**. The calc-alkaline to alkaline affinity of the magmatic rocks associated with the deposits is mostly derived from ref. 7. For the deposits which were not reported by ref. 7, literature geochemical data from previous studies (Supplementary Data 1) were used as far as possible to infer magmatic affinity using the $K_2O$ vs. $SiO_2$ plot[16]. This plot allows the discrimination of rocks into calc-alkaline, high-K calc-alkaline and alkaline (shoshonitic). For the remaining deposits for which geochemical data on the associated magmatic rocks were not available after a literature survey, discrimination was done using the nomenclature of associated porphyritic rocks provided by http://mrdata.usgs.gov/porcu/. A calc-alkaline affinity was assigned when magmatic rocks associated with the deposit were classified[48] as granites, granodiorites, tonalites, diorites, quartz-syenite, quartz-monzonite, quartz-monzodiorite (and/or their effusive equivalents: rhyolites, rhyodacites, dacites and andesites). A mildly alkaline to alkaline affinity was assigned when magmatic rocks associated with the deposit were classified[48] as (foid-bearing or not) syenites, monzonites and monzodiorites (and/or their effusive equivalents: trachytes and latites).

## Data availability

All data generated and analysed during this study are included in this published article and its Supplementary Information (Supplementary Notes 1 and 2, Supplementary Figs. 1–5, Supplementary Tables 1–6) and Supplementary Datas 1 and 2. The source data underlying Figs. 1–5 and Supplementary Fig. 1 are provided as Supplementary Datas 1 and 2.

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

## Acknowledgements

I would like to thank Lluís Fontboté for precious advices and discussions. This study was funded by the Swiss National Foundation (grant No. 200021_169032).

## Author contributions

M.C. designed the research, performed the Monte Carlo simulations, wrote the paper and drafted the figures.

## Competing interests

The author declares no competing interests.
