## [Peer Review File · Nature Communications]

Reviewers' comments:

Reviewer #1 (Remarks to the Author):

The manuscript relates the Cu versus Au dominance of porphyry deposits to tectonic setting, magma chemistry and formational depth and concludes that gold dominance is not controlled magmatically but by more efficient gold precipitation mechanisms. The approach adopted is novel and the conclusions are worthy of publication and further testing.

A few comments, mainly grammatical in nature, are marked in red on the scanned manuscript. Numbers 1-5 in the left margin refer to the comments below, which the author might consider before submission of the final manuscript.

1. Although reviewers are not supposed to draw attention to their own work these days, it would be nice to make reference to the first recognition of porphyry deposits in post-collisional settings (nine years before Richards drew renewed attention to it); Sillitoe, R. H., Gold-rich porphyry deposits: Descriptive and genetic models and their role in exploration and discovery. *Rev. Econ. Geol.*, 13, 315-345 (2000).

2. If it's an Au-rich deposit it must be enriched in gold!

3. And many other Au-rich porphyries with low- to medium-K calc-alkaline rocks: Pampa Escondida, Esperanza and Caspiche in Chile, Cascabel and Mirador in Ecuador, La Colosa, Murindó and Nuevo Chaquiro in Colombia, and Cobre Panama in Panama to name just a few not mentioned although possibly considered in Fig. 1. In fact, there are many gold-rich porphyry copper and porphyry gold-only deposits that are not associated with weakly or strongly alkaline rocks (possibly the majority).

4. But isn't it obvious that gold-poor deposits will have a lower rate of gold deposition than gold-rich deposits, assuming all else is equal, just because there's less gold deposited. If the copper-rich deposits take longer to form, as is shown, then their gold deposition rate is slower still.

5. But some of the deposits listed above, most notably Pampa Escondida and Esperanza in Chile, were not formed in island arcs with a thin crust but in thickened Andean crust that would appear to be the same thickness as that present during nearby Au-poor porphyry copper deposit generation. This fact needs to be addressed as does the case of Grasberg, which is a composite deposit formed by juxtaposition of the world's most gold-rich porphyry deposit and a gold-deficient porphyry copper-molybdenum deposit. The gold trend in Fig. 1 is heavily dependent on Grasberg, which is in fact two very different deposits side-by-side.

6. The trends in Fig. 1 seem heavily dependent on giant deposits that are in fact composite and comprise several discrete deposits in close proximity (see comments on Grasberg, above). El Teniente may be the only one of these giants that can be considered a single deposit. Does this bias the conclusions?

Comment: Au-rich porphyry deposits, as noted in the manuscript, are formed more shallowly than porphyry copper-molybdenum deposits. Since many weakly to strongly alkaline magmas are emplaced in extensional settings, couldn't that explain why they typically give rise to shallow deposits? If so, then emplacement depth could be the prime explanation for Au enrichment, without the need to invoke tectonic setting or other parameters considered in the manuscript. Worth thinking about.

Richard Sillitoe

Reviewer #2 (Remarks to the Author):

Interesting paper that is worthy of publication. The basic tenet here is that the tonnage of a gold or copper ore deposit depends primarily on duration of mineralization, which the author attributes to differences in the size of magmatic systems and the depths of their intrusion.

The data are interesting and, as far as I can tell, the first time these relationships have been shown. No doubt, these relationships will spawn discussions in the community.

There are however some ways the paper could be improved for NC.

1) There should probably be more discussion about mechanisms of magma transport, precipitation of Au or Cu, the role of tectonics in controlling magmatic flux, etc.

2) Monte Carlo simulations are somewhat opaque. In NC, the authors have more room to discuss, so it would be wise to lay out the details in the text.

3) Figure 1 is difficult to understand because the gray symbols reflect MC model outputs. The author needs to explain in more detail what is involved in these calculations.

4) Some discussion on the methods and uncertainties of estimating mineralization duration is warranted. Maybe even some discussion on what length of ore formation actually means or how it is defined.

5) Need to define better to the reader what calc-alkaline and high K-calc-alkaline means and what criteria were used to categorize magmas as such. I think this is important if the author wants this work to be read by people outside of the porphyry business.

6) Model results in Fig 2 are not obvious from figure. Some of this may be in supplemental, but if the reader can't get the gist from the Figure, details must be laid out in the main text or figure caption or both. What were the parameters for the MC simulation? How do you calculate total available fluids? What assumptions were made on the background Au (or Cu) concentrations?

Reviewer #1 (Remarks to the Author):

The manuscript relates the Cu versus Au dominance of porphyry deposits to tectonic setting, magma chemistry and formational depth and concludes that gold dominance is not controlled magmatically but by more efficient gold precipitation mechanisms. The approach adopted is novel and the conclusions are worthy of publication and further testing.

A few comments, mainly grammatical in nature, are marked in red on the scanned manuscript. Numbers 1-5 in the left margin refer to the comments below, which the author might consider before submission of the final manuscript.

1. Although reviewers are not supposed to draw attention to their own work these days, it would be nice to make reference to the first recognition of porphyry deposits in post-collisional settings (nine years before Richards drew renewed attention to it); Sillitoe, R. H., Gold-rich porphyry deposits: Descriptive and genetic models and their role in exploration and discovery. *Rev. Econ. Geol.*, 13, 315-345 (2000).

Thanks. I acknowledge this remark and I have added the reference.

2. If it's an Au-rich deposit it must be enriched in gold!

This has been corrected as suggested by the reviewer (line 37).

3. And many other Au-rich porphyries with low- to medium-K calc-alkaline rocks: Pampa Escondida, Esperanza and Caspiche in Chile, Cascabel and Mirador in Ecuador, La Colosa, Murindó and Nuevo Chaquiro in Colombia, and Cobre Panama in Panama to name just a few not mentioned although

possibly considered in Fig. 1. In fact, there are many gold-rich porphyry copper and porphyry gold-only deposits that are not associated with weakly or strongly alkaline rocks (possibly the majority).

Yes this is true. I have updated the database with the additional deposits suggested by the reviewer and also others that were reported by Murakami et al. (2010). The database (Table S1) now is more complete (118 deposits) and the arguments made are reinforced since the newly added deposits keep following the same trends (Au-rich versus Cu-rich) outlined by the previous database.

4. But isn't it obvious that gold-poor deposits will have a lower rate of gold deposition than gold-rich deposits, assuming all else is equal, just because there's less gold deposited. If the copper-rich deposits take longer to form, as is shown, then their gold deposition rate is slower still.

Not entirely, because gold-poor deposits could form at the same rate as gold-rich ones but during an overall shorter time interval: in this case both Au-rich and Au-poor deposits would fall on the same linear trend in the Au-time plot, but Au-poor deposits would fall at the low end and Au-rich at the high end of the trend. However, in this case Au-rich and Au-poor (=Cu-rich) deposits fall on two distinct trends in the Au-time plot suggesting two distinct average rates of Au precipitation. I gave values to have an idea of the order of magnitude of the difference of average rate of Au deposition between the two types.

5. But some of the deposits listed above, most notably Pampa Escondida and Esperanza in Chile, were not formed in island arcs with a thin crust but in thickened Andean crust that would appear to be the same thickness as that present during nearby Au-poor porphyry copper deposit generation. This fact needs to be addressed as does the case of Grasberg, which is a composite deposit formed by juxtaposition of the world's most gold-rich porphyry deposit and a gold-deficient porphyry copper-molybdenum deposit. The gold trend in Fig. 1 is heavily dependent on Grasberg, which is in fact two very different deposits side-by-side.

Thanks for the observation. I have now added also Pampa Escondida for which I found Cu and Au data in the BHP Annual Report 2017. It is indeed a deposit that contains a lower Cu/Au ratio (i.e., higher Au) than the neighbor Escondida (for which there is no Au data in the report, although apparently gold is produced as by-product). Nonetheless, Pampa Escondida falls along the broad trend of Cu/Au versus depth plot (new Fig. 5a-b reported also below with the identification of Pampa Escondida). In this sense it does not seem to represent an exception. In terms of contained gold according to BHP annual report 2017 Pampa Escondida (135.6 tons Au) is much lower with respect to the giant Au-rich deposit, also consistent with its deep formation. It remains the fact that, as highlighted by the reviewer, the coeval occurrence of these 2 nearby deposits (Escondida and Pampa) with apparently largely different Cu/Au deserves further investigation which is beyond the scope of this study.

Concerning Grasberg, from data reported by Wafforn (2017) it would seem that the Cu/Au ratio is very similar in the different mineralized bodies as summarized below, so it would seem that the two metals were introduced in a similar ratio throughout the mineralization period (perhaps I am missing something?):

Mine	Cu tons	Au tons	Cu/Au molar	Log Cu/Au molar
Grasberg PCD	3.16E+10	2.90E+06	3.38E+04	4.53
Big Gossan	2.26E+09	1.25E+05	5.60E+04	4.75

skarn				
Ertsberg-				
Grasberg	9.53E+10	8.18E+06	3.61E+04	4.56

An update of Au endowments according to the usgs site shows that also Pebble falls very close to Grasberg lending further support to the trend (Fig. 1).

6. The trends in Fig. 1 seem heavily dependent on giant deposits that are in fact composite and comprise several discrete deposits in close proximity (see comments on Grasberg, above). El Teniente may be the only one of these giants that can be considered a single deposit. Does this bias the conclusions?

This is true. I think that the model is consistent with the poly-phase nature of the largest deposits which have both longer time intervals of formation and probably, as highlighted by the reviewer, also larger “spatial” (composite) distribution. So I would say that this does not bias the conclusions.

Comment: Au-rich porphyry deposits, as noted in the manuscript, are formed more shallowly than porphyry copper-molybdenum deposits. Since many weakly to strongly alkaline magmas are emplaced in extensional settings, couldn't that explain why they typically give rise to shallow deposits? If so, then emplacement depth could be the prime explanation for Au enrichment, without the need to invoke tectonic setting or other parameters considered in the manuscript. Worth thinking about.

Richard Sillitoe

I perfectly agree with the reviewer's point. In the revised version I have discussed more in detail this point (lines 213-216 and 340-400) and added Fig. 4b that shows how modeling can explain the formation of Au-rich deposits in association with calc-alkaline magmas. On the other hand, the correlation existing between Sr/Y of magmatic rocks (~depth of magma evolution) and depth of porphyry formation (new Fig. 5c reported also below; only exception is Chino/Santa Rita falling outside the trend) suggests that there is a correlation between magma evolution depth and depth of magma emplacement, which is probably best explained by tectonic ± crustal

thickness.

All editorial and grammar corrections made by the reviewer #1 on the pdf file have been implemented in the revised version.

Reviewer #2 (Remarks to the Author):

Interesting paper that is worthy of publication. The basic tenet here is that the tonnage of a gold or copper ore deposit depends primarily on duration of mineralization, which the author attributes to differences in the size of magmatic systems and the depths of their intrusion.

The data are interesting and, as far as I can tell, the first time these relationships have been shown. No doubt, these relationships will spawn discussions in the community.

There are however some ways the paper could be improved for NC.

1) There should probably be more discussion about mechanisms of magma transport, precipitation of Au or Cu, the role of tectonics in controlling magmatic flux, etc.

The Discussion has been expanded including statements about magma transport and the role of tectonics in controlling magma fluxes (lines 348-350) as well as on the precipitation of Au or Cu (lines 273-310).

2) Monte Carlo simulations are somewhat opaque. In NC, the authors have more room to discuss, so it would be wise to lay out the details in the text.

I understand this point. In view of the larger allowance of space I have included now a conceptual explanation of the model in the main text (lines 46-132), a detailed explanation of the model rationale including equations in the Methods (lines 472-608), a new Table (Table 1) reporting the input parameter ranges, and additional information in the Supplementary Information.

3) Figure 1 is difficult to understand because the gray symbols reflect MC model outputs. The author needs to explain in more detail what is involved in these calculations.

The gray symbols have been removed from Fig. 1a and replotted in Fig. 4a to better subdivide “real” data from simulations. This allows a more fluent

discussion and allows a better reading of Fig. 1a. Additionally, Fig. 4b reports now also simulations for Au-rich porphyry deposits associated with calc-alkaline magmas. In view of the expanded information on the model explained above now the details of the calculations should be clear.

4) Some discussion on the methods and uncertainties of estimating mineralization duration is warranted. Maybe even some discussion on what length of ore formation actually means or how it is defined.

This is an important point. I have explained in a general way the meaning of the duration of the ore formation process in the main text (lines 75-97) and added a detailed description of how such a value was obtained for each deposit in the Supplementary Information (Details on the calculations of the duration of ore process).

5) Need to define better to the reader what calc-alkaline and high K-calc-alkaline means and what criteria were used to categorize magmas as such. I think this is important if the author wants this work to be read by people outside of the porphyry business.

This has been done both in main text (lines 61-68), and in a more detailed way in the Supplementary Information (Magmatic affinity (calc-alkaline versus high-K calc-alkaline to alkaline)).

6) Model results in Fig 2 are not obvious from figure. Some of this may be in supplemental, but if the reader can't get the gist from the Figure, details must be laid out in the main text or figure caption or both. What were the parameters for the MC simulation? How do you calculate total available fluids? What assumptions were made on the background Au (or Cu) concentrations?

This has also been done. In the main text (lines 129-132) and in Methods it is explained how exsolvable gold and copper are calculated (lines 560-588) as well as how the volume of hydrous melt is calculated (lines 485-517). Table 1 now provides the main parameters used in the model and how they have been allowed to range in the Monte Carlo simulations. This should now make clear the reading of the Figure.

REVIEWERS' COMMENTS:

Reviewer #1 (Remarks to the Author):

This expanded version of the manuscript clarifies the points raised in my original review as well as making your methods and conclusions far easier to understand.

However, the question of the porphyry Cu-Au and Cu-Mo deposits side-by-side at Grasberg remains unaddressed. Big Gossan is a Cu-Au skarn alongside Grasberg and is not the deposit I was referring to. Unfortunately, documentation of the two deposits remains unpublished, although you could contact the chief geologist, Clyde Leys (Clyde.Leys@fmi.com), to see if he's prepared to let you have a copy of a manuscript that has been accepted for publication.

Richard Sillitoe

Reviewer #2 (Remarks to the Author):

revisions look fine

Answer to reviewer #1 comment

Reviewer #1 (Remarks to the Author):

This expanded version of the manuscript clarifies the points raised in my original review as well as making your methods and conclusions far easier to understand.

However, the question of the porphyry Cu-Au and Cu-Mo deposits side-by-side at Grasberg remains unaddressed. Big Gossan is a Cu-Au skarn alongside Grasberg and is not the deposit I was referring to. Unfortunately, documentation of the two deposits remains unpublished, although you could contact the chief geologist, Clyde Leys (Clyde.Leys@fmi.com), to see if he's prepared to let you have a copy of a manuscript that has been accepted for publication.

Richard Sillitoe

Thanks for the comment. I have contacted Clyde Leys as suggested but received no answer. Therefore, I have re-examined in detail Au and Cu grades of the various mineralized bodies of the Ertsberg-Grasberg district reported in Leys et al. (2012).

All Cu/Au ratios of the various bodies (Ertsberg East, Ertsberg, Dom, Big Gossan, Ertsberg intrusion-related, Kucing Liar, Grasberg (including Main Grasberg and Dalam), Grasberg-intrusion-related and bulk Grasberg = all the ore bodies above) plot along the Au-rich trend (Figure below) although, as highlighted by the reviewer, there are two bodies, which display a slightly higher Cu/Au ratio. These are the relatively "small", 361 Mt Ertsberg body (compared to a total of >4 Gt this is <9%) with a Cu/Au molar ratio of $6.39e+04$ (compared to the bulk Grasberg-Erstsberg average of $3.34e+04$) and a part of the Grasberg mineralization associated with the Kalam porphyry. This latter mineralization displays Au/Cu ratios down to ~ 0.5 (Leys et al., 2012) equivalent to Cu/Au molar ratios of $6.2e+04$. This part of the Grasberg body also displays the highest Mo contents. I think that reviewer #1 refers to this as the porphyry Cu-Mo. According to Leys et al. (2012) the increase in Cu content in this part of the Grasberg system is due to the overprinting of the Cu-Au mineralization associated with the Main Grasberg intrusion on top of the Cu(-Mo) mineralization associated with the Dalam diorite. This has resulted in a higher Cu content in some portions close to the Dalam diorite. Additionally here and also at Ertsberg higher Mo contents are associated with deeper levels which would be consistent with these systems being formed at deeper levels than the bulk of the Cu-Au mineralization at Grasberg. Therefore, the somewhat higher Cu/Au of the Dalam rocks and Ertsberg is likely a combination of a deeper level of emplacement plus an overprint of a second mineralization. Nonetheless, it must be said the even the Cu/Au ratios of the Dalam part of the

Grasberg ore body (2-stage overprinted) and of Ertsberg remain on the Au-rich side of the trends (see picture below) in line with the overall Au-rich nature of the Grasberg-Ertsberg complex.

I have added a sentence in the main text (lines 76-86) specifying that the Cu/Au ratios in multiphase systems like Grasberg may change due to changing levels of mineralization and overprinting but that I considered the bulk of the different ore bodies because they represent by far the main mineralizing style of the deposit.

Plot showing the range of all Grasberg orebodies. The dark blue field represents all ore bodies except Dalam rocks (not a proper ore body according to Leys et al., 2012, but a part of the Grasberg ore body) and the relatively small (361 Mt) Ertsberg body, which are represented by the blue line. For reference are also shown the Au-rich trend line (yellow) of $Au/Cu = 80 \cdot 10^{-6}$ and the Cu-rich trend line (red) of $Au/Cu = 4 \cdot 10^{-6}$ reported in Figure 1a.